# Preliminary Evaluation of Four Legume and Grass Species to Compose Roadside Revegetation in Piauí, Brazil

**Andressa Ribeiro** [1], **Ricardo Loiola Edvan** [2,*] , **Layne da Silva Vieira** [1], **Keurin Terezinha Bezerra Roder** [1], **Dhiéssica Morgana Alves Barros** [1], **André Pereira Batista** [1], **Rodolfo Molinário de Souza** [1], **Vanessa Paraguai** [3], **Emídio Neves de Moraes** [4] and **Antonio Carlos Ferraz Filho** [1]

[1] Campus Professora Cinobelina Elvas, Universidade Federal do Piauí, Bom Jesus 64900-000, PI, Brazil; andressa.florestal@ufpi.edu.br (A.R.); laynevieira@ufpi.edu.br (L.d.S.V.); morganabarros1@ufpi.edu.br (D.M.A.B.); apb@ufpi.edu.br (A.P.B.); rodolfosouza@ufpi.edu.br (R.M.d.S.); acferrazfilho@ufpi.edu.br (A.C.F.F.)
[2] Campus Ministro Petrônio Portela, Universidade Federal do Piauí, Teresina 64049-550, PI, Brazil
[3] CS Grãos do Piauí S.A, BR 135 km 148, Bom Jesus 64900-000, PI, Brazil; paraguai.vanessa19@gmail.com
[4] Ambientare—Soluções em Meio Ambiente, CLNW 10/11 Bl. D Ap. 107 Noroeste, Brasília 70686-620, DF, Brazil; emidio.neves.moraes@gmail.com
* Correspondence: edvan@ufpi.edu.br; Tel.: +55-89-999424042

**Abstract:** A trial was conducted to investigate the growth and production characteristics of four plant species, marking the initiation of research on roadside revegetation processes in the southern region of Piauí state, Brazil. The trial was conducted in greenhouse conditions to evaluate the response of the species—two native legumes (*Arachis pintoi* and *Stylosanthes macrocephala*) and two grasses (*Brachiaria humidicula*—non-native and *Paspalum notatum*—native)—under different fertilization and irrigation treatments. Data were collected in two harvest operations, measuring the following variables: total plant height, population density per pot, number of live leaves, plant moisture content, total forage biomass, and root biomass. The results suggested that fertilization and irrigation caused no significant effect on the major species development characteristics that allay with the highway agency interests. *Arachis pintoi* showed the best results with the lowest height (24.1 cm in Experiment 1 and 19.2 cm in Experiment 2) and the greatest total forage biomass yield (6.4 g plant$^{-1}$ in Experiment 1 and 4.1 g plant$^{-1}$ in Experiment 2). Thus, we recommend that the results found in this study should be extended to field experiments and long-term research. Because our study did not explore mixed-species designs, adopting such evaluation could offer advantages in achieving more comprehensive and resilient revegetation outcomes and help decision-making regarding target species to compose the roadside revegetation operations.

**Keywords:** highway; environment; growth and production characteristics

## 1. Introduction

Roadways are part of contemporary landscapes and are necessary to interconnect places, facilitate transportation, and provide expansion and consolidation of the agricultural frontier [1,2]. However, roads are also indicators of anthropogenic pressure and habitat loss [3,4], which highlights the importance of studies that examine the environmental and economic benefits of roadside revegetation. The ideal study requires long-term research and, according to [2], while in Brazil we have had a considerable increase in recent studies on the subject, there are large gaps in knowledge about the effects of highways, and especially on how they affect the population of animals and humans, and ways to minimize or mitigate environmental impacts.

The 'Transcerrados' Highway is partly situated in Piauí state in a location known as 'Matopiba', which is considered one of the 'final' agricultural frontiers of Brazil. This region is one of the most dynamic in the country in terms of land conversion [5]. In addition, as its

name suggests, 'Transcerrados' is inserted in the Cerrado biome, which accounts for nearly a quarter of the country's national territory and more than half of its soy production [6]. In 2021, 'Transcerrados' became part of a public concession, which was considered the largest road intervention ever conducted in Piauí State (276.8 km). The landscape transformation, particularly in areas affected by roadways, requires evaluation of vegetation design. Because each ecosystem's territory has its basic characteristics and peculiar natural environments, they must be considered and analyzed in road environmental studies, observing the vulnerability and fragility conditions of the area [7]. Thus, the selection of plant species to compose the roadside vegetation after building intervention is an important aspect of mitigating actions aimed at reducing landscape impact. The decision should be on an easy-to-manage vegetation cover that meets the specifications of highway authorities with respect to safety, erosion control, and maintenance [8], ideally allying with the interests of the highway agencies and maintainers, such as lower costs and easy management.

A well-established roadside vegetation can improve aesthetics, increase property values, reduce heat, control surface water runoff, and reduce noise pollution [9]. Species selection should consider some desired characteristics to achieve successful revegetation, such as fast establishment and growth, high germination rate, easy acquisition of seeds, rusticity, and minimal resource requirements such as water, fertilizers, and pesticides [8,10–12]. In addition, concerns regarding the threat of introduced invasive species have increased the promotion of native plants in roadside landscapes [13,14]. Non-native species are used in revegetation projects because of the ability of grasses to provide almost immediate erosion control [12,15], while native species will usually be more robust and resistant to local climatic conditions [9].

Therefore, the aim of this study was to evaluate the growth and production characteristics of four species (legumes × grasses and native × non-native) under different fertilization and irrigation. The species considered included a non-native grass (*Brachiaria humidicula* (Rendle) Schweick—commonly referred to as 'braquiária') and three other species native to Latin America: *Arachis pintoi* Krapovickas and Gregory—known as 'amendoim-forrageiro' (legume), *Stylosanthes macrocephala* Ferr. et Costa—named 'estilosantes' (legume), and the grass *Paspalum notatum* Flüggé—known as 'grama-batatais'. These species are initial candidates that meet the objectives of the highway agency to compose the roadside revegetation of the PI 262 highway, which is part of the 'Transcerrados' in Piauí state.

## 2. Materials and Methods

### 2.1. Experimental Design

This study was conducted from July to December 2022 under greenhouse conditions at the Federal University of Piauí (9°05′00″ S and 44°19′34″ W). Because this is a preliminary study and was developed within a constricted schedule and budget, we decided to limit the number of studied species and conduct the first trials in greenhouse conditions. This was done to better investigate processes that cannot be easily measured in the field, especially irrigation and soil correction.

Two experiments were installed using a completely randomized design with factorial arrangement, with five replications for Experiment 1 and three replications for Experiment 2. The factors corresponded to four plant species, two legumes—*Arachis pintoi*, and *Stylosanthes macrocephala*, and two grasses—*Brachiaria humidicula* and *Paspalum notatum*. In the first experiment, three soil correction treatments were tested: no correction (control), application of 5 g of NPK (6-24-12) and 2.5 g of limestone filler per pot, and application of 2.5 g of limestone filler per pot. Corrections of the tested soil were limited to a few basic treatments because it is the intention of the highway agency to perform minimum soil interventions before roadside revegetation. In Experiment 2, two types of irrigation were evaluated, 80% and 40% of the field capacity, with no soil correction. In this experiment, the control treatment (no irrigation) was not included, considering that the period of experiment implementation coincided with the dry season of the region (July), and previous

experiences in this region's prolonged dry season showed that no germination occurs as well as high mortality due to the lack of irrigation.

The pot capacity method described by [16] was applied to determine the volume of irrigation. This method consists of filling the pots with soil, obtaining each pot's weight, and then placing them in a tray with water volume equivalent to one-third of the height of the pots for a period of 24 h so that complete saturation occurs (saturation via capillarity). After this process, the pots were placed on a bench to drain the excess water for a period of 24 h and were weighed again. The difference between the two weights was considered to be equivalent to the water retained in the pot after the drainage period and was considered 100% of the pot's field capacity. After determining the pot's field capacity, irrigation was performed to reach levels of 80% and 40% of the field capacity.

The superficial layer (0–0.2 m) of a yellow dystrophic latosol with a sandy loam texture was used to fill the pots. The soil used in the experiments was collected from an area located on the PI 262 highway to simulate field conditions (9°16′56″ S and 44°50′21″ W). Even though this is not enough to reproduce all natural field conditions because other local variables could affect plant growth and production, the use of the soil from the proposed revegetation area is appropriate, considering the preliminary nature of this study. Samples representing this material were mixed to form a composite sample and subsequently taken to the laboratory for granulometric and chemical characterization, according to the methodologies recommended by [17]. The following contents were obtained: 581, 2, and 416 g kg$^{-1}$ of sand, silt, and clay, respectively. The chemical results were: 5.3 pH in water, 0.15 cmol dm$^{-3}$ of calcium, 0.08 cmol dm$^{-3}$ of magnesium, 0.04 cmol dm$^{-3}$ of potassium, 0.5 cmol dm$^{-3}$ of aluminum, 1.5 mg dm$^{-3}$ of phosphorus, 0.5% of organic matter, 2.6 cmol of cation exchange capacity, and 9.6% of base saturation.

### 2.2. Sowing and Data Collection

Sowing was conducted by placing 20 seeds per pot, followed by thinning at 20 days after emergence, leaving the three most vigorous plants per pot. Uniformization pruning occurred 70 days after sowing. Throughout the experimental period, two evaluation harvests were performed every 30 days of growth. The cuts were made at a 10 cm height above the soil surface.

Growth and production characteristics were evaluated for each cut (two harvests), and the mean values were statistically analyzed. The following data were collected: the total plant height, which was measured from the soil to the last expanded leaf (PH in cm) using a graduated ruler; population density per pot (PD) by counting the number of live tillers for grasses and sprouts for legumes present in each pot; the number of live leaves (NL); plant moisture (PM in %), which was obtained by dividing the difference between fresh and dry biomass weight by the total fresh biomass weight; total biomass (TB in g plant$^{-1}$), which was measured from samples collected in the pot; and root biomass from the last harvest (RB in g plant$^{-1}$). A digital scale with a precision of 0.01 g was used to determine the fresh plant biomass. Plant dry matter content determination (method INCT-CA G-003/1) was carried out according to methods described by [18], from which it was possible to determine the dry biomass of the plants.

### 2.3. Data Analysis

The data were subjected to analysis of variance (ANOVA), and when the treatment effects were significant, they were compared by the Scott–Knott test (*p*-value $\leq$ 0.05) using the statistical analysis system SISVAR, version 5.0 [19]. The standard error of the mean (SEM) was also calculated by taking the standard deviation value and dividing it by the square root of the sample size.

All analyzed variables were ranked according to desirable characteristics proposed by the highway agency to guide the choice of the best potential species (e.g., rapid development, easy and cost-effective establishment, low maintenance requirements, and minimal soil conditions demand). Hence, considering plant height (PH), the ideal is that the species

present lower stature so that species with lower height receive the lowest value (1) and the tallest one receive the highest value (4). For the variables population density (PD), the number of live leaves (NL), plant moisture (PM), total biomass (TB), and root biomass (RB), the desired behavior is that the species present higher values and in this way avoids soil erosion so that species with higher values received the lowest value (1) and species with lower values received the highest value (4). Species with the highest performance for a specific characteristic received a score of 1, the second-best species received a score of 2, and so on, considering, as the most desirable species, the ones with the lowest sum.

## 3. Results

The species with the fastest germination was *Arachis pintoi*, emerging about seven days after sowing, while *Paspalum notatum* presented the slowest germination, 11 days after sowing. The species *Stylosanthes macrocephala* and *Brachiaria humidicula* presented germination 9 days after sowing. Data for the studied variables for both experiments are presented in Figures 1 and 2, where the data dispersion was greater in Experiment 1 when compared with Experiment 2.

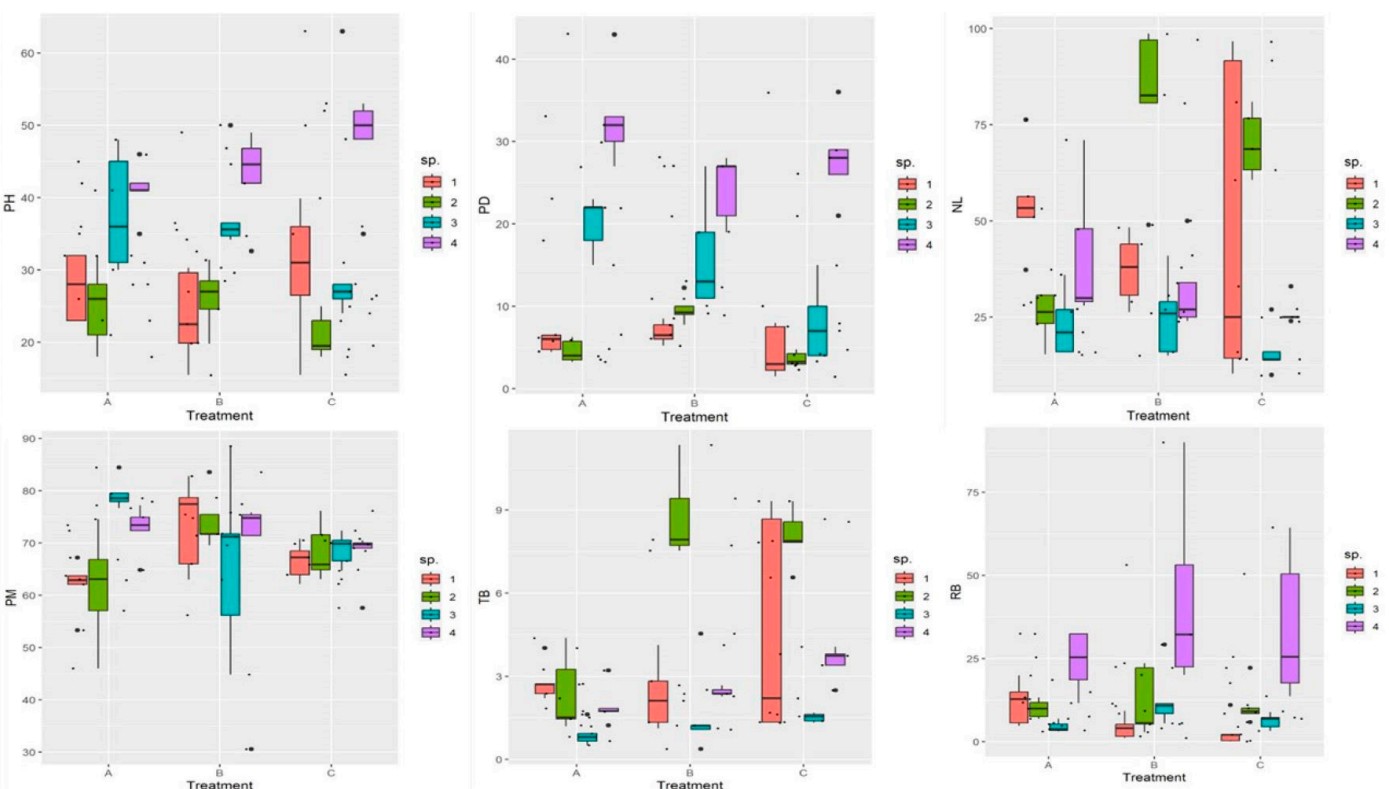

**Figure 1.** Box plot and dispersion of Experiment 1 for total height of the plant (PH in cm), population density per pot (PD), number of live leaves (NL), plant moisture (PM in %), total biomass (TB in g plant$^{-1}$), and root biomass (RB in g plant$^{-1}$). Where species (sp.) is 1 = *Stylosanthes macrocephala*, 2 = *Arachis pintoi*, 3 = *Brachiaria humidicula*, and 4 = *Paspalum notatum*, and treatment is A = control, B = soil + NPK + limestone filler, and C = soil + limestone filler.

Except for plant moisture (PM), which did not have a significant effect on any of the factors, all the other variables show individual effects of the plant species ($p < 0.01$), whereas fertilization had a significant effect only on the population density per pot—PD ($p = 0.04$) and total biomass—TB ($p < 0.01$). No significant effect of the interaction between plant species and fertilization is found on any of the evaluated variables ($p > 0.05$) (Table 1).

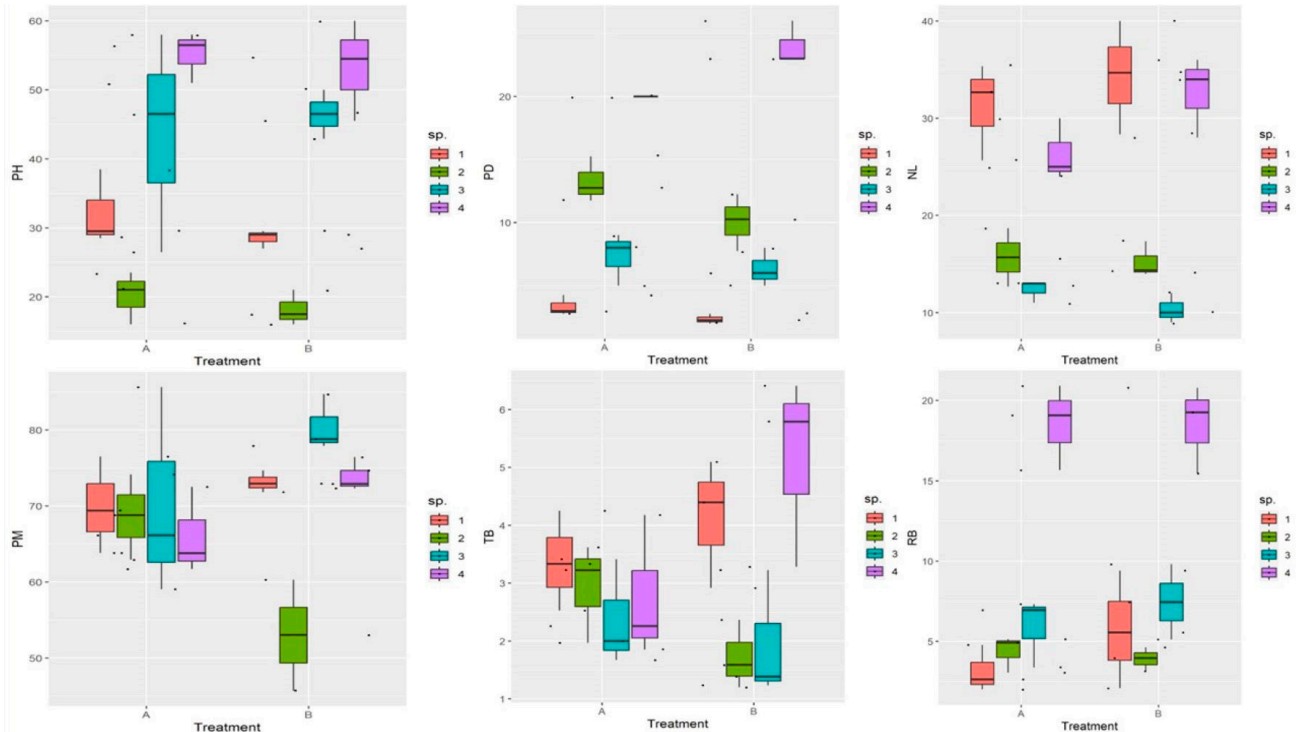

**Figure 2.** Box plot and dispersion of Experiment 2 for total height of the plant (PH in cm), population density per pot (PD), number of live leaves (NL), plant moisture (PM in %), total biomass (TB in g plant$^{-1}$), and root biomass (RB in g plant$^{-1}$). Where species (sp.) is 1 = *Stylosanthes macrocephala*, 2 = *Arachis pintoi*, 3 = *Brachiaria humidicula*, and 4 = *Paspalum notatum*, and treatment is A = 80% and B = 40% of the field capacity.

In the second experiment, no significant effect of the interaction between plant species and irrigation was found on any of the evaluated variables ($p > 0.05$). However, all variables showed individual effects of the plant species ($p < 0.01$), while the effect of irrigation was not significant on any of the evaluated variables ($p > 0.05$, Table 2). It should be noted that mites attacked the *Arachis pintoi* pots, which may have contributed to its low biomass production.

**Table 1.** Analysis of variance of Experiment 1, effect of fertilization and potential species for roadside revegetation in Piauí, Brazil.

| Variables * | Plant Species ** | | | |
|---|---|---|---|---|
| | *Arachis pintoi* | *Stylosanthes macrocephala* | *Brachiaria humidicula* | *Paspalum notatum* |
| PH (cm) | 24.1 [c] | 27.0 [c] | 43.9 [a] | 36.6 [b] |
| PD | 5.9 [c] | 5.6 [c] | 28.5 [a] | 14.7 [b] |
| NL | 59.0 [a] | 46.6 [a] | 33.2 [b] | 21.6 [b] |
| PM (%) | 68.1 | 67.3 | 68.0 | 71.6 |
| TB (g plant$^{-1}$) | 6.4 [a] | 3.2 [b] | 2.6 [b] | 1.4 [c] |
| RB (g plant$^{-1}$) | 11.0 [b] | 6.3 [b] | 7.9 [b] | 34.0 [a] |

| Variables * | Fertilization ** | | |
|---|---|---|---|
| | Control | Limestone filler | Limestone filler + NPK |
| PH (cm) | 32.9 | 32.8 | 33.0 |
| PD | 15.8 [a] | 14.3 [a] | 11.0 [b] |
| NL | 26.1 | 44.1 | 40.1 |
| PM (%) | 68.8 | 70.0 | 67.7 |
| TB (g plant$^{-1}$) | 2.1 [b] | 3.8 [a] | 4.4 [a] |
| RB (g plant$^{-1}$) | 12.5 | 18.2 | 13.7 |

**Table 1.** *Cont.*

| Variables * | Plant Species ** | | | |
|---|---|---|---|---|
| | *Arachis pintoi* | *Stylosanthes macrocephala* | *Brachiaria humidicula* | *Paspalum notatum* |
| Variables * | *p*-Value | | | SME *** |
| | Plant species | Fertilization | Species × Fertilization | |
| PH (cm) | <0.01 | 0.99 | 0.45 | 2.03 |
| PD | <0.01 | 0.04 | 0.052 | 0.95 |
| NL | <0.01 | 0.33 | 0.12 | 4.34 |
| PM (%) | 0.60 | 0.74 | 0.08 | 2.39 |
| TB (g plant$^{-1}$) | <0.01 | <0.01 | 0.06 | 0.39 |
| RB (g plant$^{-1}$) | <0.01 | 0.28 | 0.30 | 3.03 |

\* PH = total plant height; PD = population density per pot; NL = number of live leaves; PM = plant moisture; TB = total biomass; RB = root biomass. ** Means followed by different lowercase letters in the same row are statistically different at $p < 0.05$ according to the Scott–Knott test. *** Standard error of the mean.

**Table 2.** Analysis of variance of Experiment 2, effect of irrigation and potential species for roadside revegetation in Piauí, Brazil.

| Variables * | Plant Species ** | | | |
|---|---|---|---|---|
| | *Arachis pintoi* | *Stylosanthes macrocephala* | *Brachiaria humidicula* | *Paspalum notatum* |
| PH (cm) | 19.2 [d] | 30.3 [c] | 54.2 [a] | 45.1 [b] |
| PD | 11.6 [b] | 2.8 [d] | 22.0 [a] | 6.8 [c] |
| NL | 15.4 [b] | 32.8 [a] | 29.5 [a] | 11.3 [b] |
| PM (%) | 60.8 [b] | 71.5 [a] | 75.3 [a] | 70.0 [a] |
| TB (g plant$^{-1}$) | 2.3 [b] | 3.8 [a] | 4.0 [a] | 2.2 [b] |
| RB (g plant$^{-1}$) | 4.1 [b] | 4.4 [b] | 6.7 [b] | 18.5 [a] |
| Variables * | Irrigation ** | | | |
| | 80% of field capacity | | 40% of field capacity | |
| PH (cm) | 37.8 | | 36.6 | |
| PD | 10.9 | | 10.6 | |
| NL | 21.3 | | 23.1 | |
| PM (%) | 68.7 | | 70.1 | |
| TB (g plant$^{-1}$) | 2.9 | | 3.2 | |
| RB (g plant$^{-1}$) | 8.0 | | 8.9 | |
| Variables * | *p*-Value | | | SME *** |
| | Plant species | Irrigation | Species × Irrigation | |
| PH (cm) | <0.01 | 0.69 | 0.87 | 2.9 |
| PD | <0.01 | 0.66 | 0.12 | 0.7 |
| NL | <0.01 | 0.27 | 0.26 | 1.5 |
| PM (%) | 0.01 | 0.62 | 0.25 | 2.9 |
| TB (g plant$^{-1}$) | 0.01 | 0.37 | 0.06 | 0.4 |
| RB (g plant$^{-1}$) | <0.01 | 0.33 | 0.61 | 0.9 |

\* PH = total plant height; PD = population density per pot; NL = number of live leaves; PM = plant moisture; TB = total biomass; RB = root biomass. ** Means followed by different lowercase letters in the same row are statistically different at $p < 0.05$ according to the Scott–Knott test. *** Standard error of the mean.

For the purpose of a decision among the evaluated species, the ranking of the studied variables was conducted for both experiments (Table 3).

**Table 3.** Ranking based in the lowest sum of evaluated characteristics for the studied species, where PH = total height of the plant (cm), PD = population density per pot, NL = number of live leaves, PM = plant moisture (%), TB = total biomass (g/plant), and RB = root biomass (g/plant). For PH, the lowest value is attributed to the shortest species, and for PD, NL, PM, TB, and RB, the lowest value is attributed to the greatest value by species.

| Experiment | Species | PH | PD | NL | PM | TB | RB | Sum |
|---|---|---|---|---|---|---|---|---|
| 1 | *Arachis pintoi* | 1 | 3 | 1 | 2 | 1 | 2 | 10 |
| | *Stylosanthes macrocephala* | 2 | 4 | 2 | 4 | 2 | 4 | 18 |
| | *Brachiaria humidicula* | 4 | 1 | 3 | 3 | 3 | 3 | 17 |
| | *Paspalum notatum* | 3 | 2 | 4 | 1 | 4 | 1 | 15 |
| 2 | *Arachis pintoi* | 1 | 2 | 3 | 4 | 3 | 4 | 17 |
| | *Stylosanthes macrocephala* | 2 | 4 | 1 | 2 | 2 | 3 | 14 |
| | *Brachiaria humidicula* | 4 | 1 | 2 | 1 | 1 | 2 | 11 |
| | *Paspalum notatum* | 3 | 3 | 4 | 3 | 4 | 1 | 18 |

## 4. Discussion

This study is unprecedented, representing the first step towards the selection of potential species for revegetation of the roadside areas of 'Transcerrados'. According to the instructions of the National Highway Agency [20], the selection of plant species for roadside revegetation should be focused on their self-sustainability within their ecological community, considering their role in maintaining the local fauna. The highway concessionaires aim to revegetate the roadsides with species that exhibit desirable characteristics, such as rapid development, easy and cost-effective establishment, low maintenance requirements, and minimal soil condition demands. Ideally, the vegetation for roadside areas should provide adequate ground coverage and present a short height.

Following these guidelines and evaluating only the treatment without any soil correction in Experiment 1 (Table 1), as it represents the ideal condition for the concessionaire, the species that produced the most biomass was *Arachis pintoi*, while the species with the lowest biomass yield was *Paspalum notatum*. In Experiment 2 (Table 2), the ideal species would be the one that tolerates water stress, considering that no irrigation system is planned for the roadside revegetation. *Brachiaria humidicula* was the species that produced the most biomass, and once again, *Paspalum notatum* showed the lowest biomass yield. Another crucial point to be evaluated is the height of the vegetation cover, as Brazilian legislation (ANTT 4.071/2013) mandates that the roadside areas should have vegetation heights lower than 30 cm. Therefore, both legume species exhibited the shortest statures in both experiments, with *Stylosanthes macrocephala* being taller than *Arachis pintoi*. The grass *Brachiaria humidicula* had the highest height, followed by *Paspalum notatum*.

It is important for the species to provide good ground coverage and adaptability to the soil and climate factors, as exemplified by *Arachis pintoi*, which is native to the region and also has a short height, ensuring good visibility on the roads and producing lesser material for potential fires when compared with other commonly used species. Native species are recommended for planting alongside the highways, mainly due to their better ecological balance with the local ecosystem, greater environmental adaptability, and their role in serving as landscape corridors [13].

A constant concern of both highway concessionaires and environmentalists is the risk of ignition and the possibility of wildfires on the roadsides. Although the presence of vegetation in these roadside areas benefits the environment, they also represent a potential ignition source of wildfires. Fire could easily spread through the roadside vegetation present in the road median and edges due to the lack of vegetation maintenance, the fuel load, and the flammability of the main species [21]. Fire regimes can be altered by introducing non-native plants when they change fuel properties, resulting in positive ecological feedback, where the invasion of a non-native plant increases fire frequency and/or intensity of fires, sometimes beyond the level where native vegetation can recover.

This phenomenon has been particularly documented in high-biomass grasses and is often referred to as the 'grass–fire cycle' [22]. It is important to highlight that analyses have indicated that non-native plant species in a region have the potential to amplify the effect of fires in vegetated areas when compared with native species [23].

The moisture content of a plant species can be an indicative factor of its potential as an ignition source, making species with higher moisture content preferable for roadside revegetation. However, the results of this study revealed no statistically significant differences in moisture content among species. In Experiment 1, the limestone filler treatment increased the moisture value, while in Experiment 2, the moisture content varied among species, with the lowest value observed in *Arachis pintoi*. Because the interest is to avoid soil correction and irrigation, the grass species had higher moisture values. However, it is important to note that these grasses also produce more combustible material and present greater height, making them potentially invasive species, particularly *Brachiaria humidicula*.

Legume plants produce a significant amount of biomass that is rich in minerals. They also possess a deep and branched root system capable of extracting nutrients from deeper soil layers, which are made available after their decomposition and incorporation [24]. On the other hand, grasses have a fibrous and abundant root system that acts like a network, holding soil aggregates together, thereby making it more resistant to the impact of raindrops and erosion caused by runoff [25]. Erosion is also a major concern in the planning of roadside revegetation [26,27], so plant root systems can provide underground support and prevent shallow slope failures by increasing soil strength through reinforcement [27]. In the present study, *Paspalum notatum* was the only species that showed significant differences from the others regarding the root biomass yield. However, when considering the overall characteristics (Table 3), this particular species was poorly ranked when compared with *Arachis pintoi* in Experiment 1.

The value of roadside vegetation as a habitat for pollinators has gained increased attention, particularly in areas dominated by agriculture, as is the case of 'Transcerrados'. However, many factors, including safety, cost, public perception, erosion control, and weedy plants, must be considered when managing roadside vegetation [28]; thus, it should be highlighted that the importance of research developments take other variables into account for roadside revegetation projects in the region. Reference [29] reported challenges to domesticating native forage legumes in the USA, with the seed cost being a limit to extensive use, suggesting that a commercially viable seed industry to support the widespread use of native legumes will require acceptance by end users of broadly adapted, genetically diverse, and superior genotypes rather than only local ecotypes. Several biotic, edaphic, and microclimatic factors function as strong environmental filters that hinder the establishment of target plant species on roadsides [30].

Our results suggest that *Arachis pintoi* has the potential to be used for roadside revegetation because it presents good soil coverage, forming a dense layer of rooted stolons that provide effective protection against erosive effects caused by rainfall, making it an excellent choice for roadside revegetation. Additionally, it has the advantage of not having a climbing growth habit, which reduces maintenance costs [31,32]. Moreover, it serves as a forage species with nitrogen fixation capabilities [33] and is attractive to wildlife, especially pollinators, making it beneficial for gardens, erosion control, and easily adaptable to natural environments [34]. On the other hand, *Brachiaria humidicula* is not a native plant species, and despite being better ranked in Experiment 2 (Table 3), it has a taller growth potential, reaching up to one meter [35], with potential of invasiveness, which could cause environmental imbalances [36].

An important aspect to consider in the selection of plant species for roadside revegetation is the cost of seed acquisition. In this study, the cost of seeds for the studied species followed the order: *Arachis pintoi* (337.50 BRL kg$^{-1}$) > *Brachiaria humidicula* (116.10 BRL kg$^{-1}$) > *Paspalum notatum* (62.10 BRL kg$^{-1}$) > *Stylosanthes macrocephala* (16.50 BRL kg$^{-1}$). This cost factor may significantly influence the decision-making process for the selection of species, particularly for large-scale revegetation projects. Because Piauí state is in the northeast of

Brazil and the major seed commerce centers are located in the south or southeast regions of Brazil, seed acquisition is challenging, with high shipping values and minimum quantity sales.

Selecting the right plant species for any purpose requires long-term research. While our study did not explore mixed-species designs, adopting such a design could offer advantages in achieving more comprehensive and resilient revegetation outcomes. As explained by [8], using a seed mix that incorporates both early- and late-successional species would result in both rapid revegetation and lasting vegetative cover for the long term. Reference [37] described that the succession of events after the revegetation of an area is generally not studied, as is the case in the present research. Over time, a plant consortium that initially seemed suitable may not be adequate in the future, leaving the area with exposed soil and vulnerable to the effects of weathering. A particular challenge for highway agencies and other land managers to follow timelines occurs when weather events wash away seeds, topsoil, fertilizers, and mulches before the vegetation takes root on the land [26].

When evaluating the soil, it is essential to take regional factors into account as well. This includes aspects such as the presence of heavy metals and salinity levels, which could pose challenges to the revegetation project. As a result, we recommend that future studies incorporate measurements of these and other soil variables to gain a comprehensive understanding of the roadside environment. It is also necessary to conduct studies on the ecological implications of roads [38], as there is an ongoing need to develop and assess mitigation measures. Therefore, we suggest that for the correct decision-making regarding roadside revegetation, continuous and broader studies should be conducted to infer the set of potential species to compose the roadside areas in southern Piauí.

## 5. Conclusions

This is a preliminary study to choose potential species that meet the highway agency goals for roadside revegetation at the PI 262 highway in Piauí, northeast of Brazil. The four studied species presented good development. *Arachis pintoi* (amendoim-forrageiro) is indicated as the most preeminent species because it presents the lowest plant height, good biomass production, and a large number of live materials, being a native legume species with high value for pollinators.

We suggest that more research be conducted to confirm our findings and to determine other possible target species (and mixed usage of species), accounting for more variables (e.g., water storage and loss, nutrient cycling, surface stability, ecological effects, fauna) to be studied in experiments conducted in field conditions, and incorporating a wider range of species to provide a more complete picture of plant responses and interactions.

**Author Contributions:** Conceptualization, A.R. and R.L.E.; methodology, A.R., R.L.E. and D.M.A.B.; software, R.L.E.; validation, A.R., R.L.E. and A.C.F.F.; investigation, R.M.d.S., A.C.F.F., D.M.A.B., R.L.E. and A.R.; resources, L.d.S.V., K.T.B.R. and A.P.B.; data curation, A.R., R.L.E., L.d.S.V., K.T.B.R., A.P.B. and D.M.A.B.; writing—original draft preparation, A.R.; writing—review and editing, A.R., R.L.E., R.M.d.S. and A.C.F.F.; supervision, A.R. and R.L.E.; project administration, A.R., V.P. and E.N.d.M.; funding acquisition, A.R., V.P. and E.N.d.M. All authors have read and agreed to the published version of the manuscript.

**Funding:** This paper is part of the project "Avaliação de espécies vegetais para compor faixas de domínio rodoviário no Piauí" (UFPI n. 23111.014806/2022-32), funded by CS Grãos do Piauí S.A.

**Data Availability Statement:** The data presented in this study are available upon request to the corresponding author.

**Acknowledgments:** We are thankful to the Empresa Florestal do Piauí Júnior (Eflopi Jr) and CS Grãos do Piauí S.A for supporting the project execution.

**Conflicts of Interest:** The authors declare no conflict of interest.

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
