# Peer review of "Preliminary Evaluation of Four Legume and Grass Species to Compose Roadside Revegetation in Piauí, Brazil"

_agronomy, doi:10.3390/agronomy13092283_

Round 1

Reviewer 1 Report

Line 75: why greenhouse conditions? How does it relate to the roadside conditions? Please explain why you opted for this kind of trial establishment instead of the open field conditions. 

Compliments on the trial establishment with native and non-native species. 

Please consider providing the full data, for each given species, not just the average when presenting the effect of fertilization and irrigation in Tables 1 and 2.  It can be a supplement to avoid overwhelming the manuscript.

Lines 163-175: What about heavy metals and salinity along the roadside? Why was it omitted from the study? Again, the greenhouse does not represent the site conditions when thinking of the roadside....

Lines 221 - 227: How about the mixed usage of species? Those that control soil and erosion, hold water, provide nutrients... In my opinion, your next trial should consider mixed sowing. 

Please provide separate Conclusion action with main findings, supported with exact measured values. 

Lines 228 - 237 are highly valuable. 

Author Response

Dear reviewer,
The suggestions were accepted, file attached with the details.

Reviewer 2 Report

The topic is suitable for the journal.

The Introduction section is clear and the aims have been clearly explained as well.

The experimental design is scientifically sound and the statistical analyses are appropriate.

Minor comments

Lines 46-48 – Modify the sentence: "Since each ecosystem’s territory has its basic characteristics and peculiar natural environments, they must be considered and analyzed in road environmental studies, observing the vulnerability and fragility conditions of the area.”

l. 91 – It is unclear to me how you determined the field capacity. The cited publications are in Portuguese; thus, it is difficult for the reader to understand the protocol. Please, explain using a few lines.

l. 113 – Describe how the leaf blade/stalk ratio is measured. Do you mean leaf blade and stalk (petiole?) length? Is there any reference that describes the method?

Explain the meaning of the variables measured in the context of your study. It is not clear for all of them. For example, it is clear for PH, TB, and PM, but what about the other variables? In other words, justify using each variable to achieve the aim of your study.

Explain in the Data analysis section the ranking used for each variable (e.g. score 1 was given to lower PH, TB, etc.) and the reasoning behind this sorting.

Table 2 - L/S. The letter “b” should be in superscript

Table 3 – As the table should be self-explanatory, explain also in the legend the ranking for each variable (e.g. score 1 was given to lower PH, TB, etc.).

References section – Translate into English the titles of publications in Portuguese and write in parentheses (in Portuguese).

Author Response

Answer Letter

Dear Editor,

We appreciate the attention and contribution of your reviewers in the analysis of the possible publication of this article. We have modified the article according to the requests with track changes control active. We have also submitted the article to an English reviewer, which improved the quality of our text. We would like to appreciate the academic editor and reviewers for the valuable suggestions that certainly improved our manuscript.

Below is the survey of the requests and answers.

Best regards,

Dr. Ricardo Loiola Edvan,

Professor at Federal University of Piauí.

Reviewer 1

Answer

Line 75: why greenhouse conditions? How does it relate to the roadside conditions? Please explain why you opted for this kind of trial establishment instead of the open field conditions.

Agreed and inserted the follow text:

“Since it was a preliminary study, we decide to start the trials in greenhouse conditions to better investigate processes that cannot be measured in the field, especially irrigation”.

Please consider providing the full data, for each given species, not just the average when presenting the effect of fertilization and irrigation in Tables 1 and 2.  It can be a supplement to avoid overwhelming the manuscript.

Agreed and figures 1 and 2 with all data dispersion was inserted for both experiments. If necessary, we can provide full data as supplement material.

Lines 163-175: What about heavy metals and salinity along the roadside? Why was it omitted from the study? Again, the greenhouse does not represent the site conditions when thinking of the roadside....

We understand and agree with the range of variables that comprehend roadside revegetation operation, but our study is a preliminary analysis so many important items that should be accounted in the definition on target species unfortunately were not included, while for lack of time or budget. We reinforce that more long-time research must be done before any major decision making. However, the soils used in the treatments were from the study area, as was affirmed in the revision. We included a sentence in the discussion about this subject.

Lines 221 - 227: How about the mixed usage of species? Those that control soil and erosion, hold water, provide nutrients... In my opinion, your next trial should consider mixed sowing.

Agreed and insert follow text:

“In this experiment, we chose not to include a control treatment (no irrigation) because the period of experiment installation coincided with the dry season and previous experiences showed no germination or high mortality without irrigation.”

We have also mentioned this in the last paragraph of the conclusions.

Please provide separate Conclusion action with main findings, supported with exact measured values.

Agreed, a conclusion was inserted.

Reviewer 2

Answer

Lines 46-48 – Modify the sentence: "Since each ecosystem’s territory has its basic characteristics and peculiar natural environments, they must be considered and analyzed in road environmental studies, observing the vulnerability and fragility conditions of the area.”

This sentence was created from the cited “Road vegetation Manual” from the Brazilian Road Agency. We did not understand if the reviewer just suggested to insert the preposition “the” before the word vulnerability or that we should rewrite the sentence.

l. 91 – It is unclear to me how you determined the field capacity. The cited publications are in Portuguese; thus, it is difficult for the reader to understand the protocol. Please, explain using a few lines.

After the citation the methodology was described. We rearranged the sentence for better understanding. Lines 91-98.

The citation also had the title translated to English.

l. 113 – Describe how the leaf blade/stalk ratio is measured. Do you mean leaf blade and stalk (petiole?) length? Is there any reference that describes the method?

Although is a common variable in forage studies, we agree to eliminate this variable in our research, since the others already provide the needed information.

Explain the meaning of the variables measured in the context of your study. It is not clear for all of them. For example, it is clear for PH, TB, and PM, but what about the other variables? In other words, justify using each variable to achieve the aim of your study.

Agreed and we better explained the chosen variable when explaining the proposed ranking.

Explain in the Data analysis section the ranking used for each variable (e.g. score 1 was given to lower PH, TB, etc.) and the reasoning behind this sorting.

Agreed and we better explained in the text.

Table 2 - L/S. The letter “b” should be in superscript

This variable was eliminated from the analysis.

Table 3 – As the table should be self-explanatory, explain also in the legend the ranking for each variable (e.g. score 1 was given to lower PH, TB, etc.).

Agreed and we better explained in the text.

References section – Translate into English the titles of publications in Portuguese and write in parentheses (in Portuguese).

Agreed and the translations were done in the text.

Round 2

Reviewer 1 Report

Thank you for implementing the required changes. 

Author Response

(The authors gave the same response as above.)
